# Genomic and Temporal Trends in Canine ExPEC Reflect Those of Human ExPEC

Paarthiphan Elankumaran,[a] Max L. Cummins,[a] Glenn F. Browning,[b] Marc S. Marenda,[b] Cameron J. Reid,[a] Steven P. Djordjevic[a]

[a]Australian Institute for Microbiology and Infection, School of Life Sciences, Faculty of Science, University of Technology Sydney, Ultimo, New South Wales, Australia
[b]Asia-Pacific Centre for Animal Health, Department of Veterinary Biosciences, Melbourne Veterinary School, Faculty of Veterinary and Agricultural Sciences, University of Melbourne, Parkville and Werribee, Victoria, Australia

**ABSTRACT** Companion animals and humans are known to share extraintestinal pathogenic *Escherichia coli* (ExPEC), but the extent of *E. coli* sequence types (STs) that cause extraintestinal diseases in dogs is not well understood. Here, we generated whole-genome sequences of 377 ExPEC collected by the University of Melbourne Veterinary Hospital from dogs over an 11-year period from 2007 to 2017. Isolates were predominantly from urogenital tract infections (219, 58.1%), but isolates from gastrointestinal specimens (51, 13.5%), general infections (72, 19.1%), and soft tissue infections (34, 9%) were also represented. A diverse collection of 53 STs were identified, with 18 of these including at least five sequences. The five most prevalent STs were ST372 (69, 18.3%), ST73 (31, 8.2%), ST127 (22, 5.8%), ST80 (19, 5.0%), and ST58 (14, 3.7%). Apart from ST372, all of these are prominent human ExPEC STs. Other common ExPEC STs identified included ST12, ST131, ST95, ST141, ST963, ST1193, ST88, and ST38. Virulence gene profiles, antimicrobial resistance carriage, and trends in plasmid carriage for specific STs were generally reflective of those seen in humans. Many of the prominent STs were observed repetitively over an 11-year time span, indicating their persistence in the dogs in the community, which is most likely driven by household sharing of *E. coli* between humans and their pets. The case of ST372 as a dominant canine lineage observed sporadically in humans is flagged for further investigation.

**IMPORTANCE** Pathogenic *E. coli* that causes extraintestinal infections (ExPEC) in humans and canines represents a significant burden in hospital and veterinary settings. Despite the obvious interrelationship between dogs and humans favoring both zoonotic and anthropozoonotic infections, whole-genome sequencing projects examining large numbers of canine-origin ExPEC are lacking. In support of anthropozoonosis, we found that most STs from canine infections are dominant human ExPEC STs (e.g., ST73, ST127, ST131) with similar genomic traits, such as plasmid carriage and virulence gene burden. In contrast, we identified ST372 as the dominant canine ST and a sporadic cause of infection in humans, supporting zoonotic transfer. Furthermore, we highlight that, as is the case in humans, STs in canine disease are consistent over time, implicating the gastrointestinal tract as the major community reservoir, which is likely augmented by exposure to human *E. coli* via shared diet and proximity.

**KEYWORDS** *Escherichia coli*, ExPEC, ST372, antimicrobial resistance, canine, dogs, genomic epidemiology, infections, one health, virulence, whole-genome sequencing

Address correspondence to Steven P. Djordjevic, Steven.Djordjevic@uts.edu.au, or Cameron J. Reid, Cameron.Reid@uts.edu.au.

The authors declare no conflict of interest.

Australia has one of the highest rates of ownership of dogs as companion animals in the world, with about 40% of households owning at least one dog (1). Most dog owners consider their dogs to be part of their family and often report high levels of physical contact with them in addition to sharing food from the household. Human-dog relationships are rooted in deep shared evolution and provide significant psychological benefits from improved self-confidence and companionship (2, 3). There is a general perception that

pets can improve their owners' health, sense of psychological well-being, and longevity (4). These benefits underpin calls to enable access of companion animals to health care facilities (5). However, there is the wide variety of pathogens that may transfer between humans and dogs, posing health risks to both (6). The benefits of dog ownership must therefore be weighed with the possible zoonotic disease implications that may be associated with these relationships (7–9).

Mammalian, avian, and reptilian species are colonized by commensal lineages of *Escherichia coli* that perform important functions in the gut. However, some *E. coli* lineages are known to cause severe intestinal and extraintestinal disease. Extraintestinal pathogenic *E. coli* (ExPEC) is the most common cause of Gram-negative infections in humans, causing a wide range of afflictions, including lower urinary tract infections, pyelonephritis, bacteremia, sepsis, skin infections, and ventilator-associated respiratory infections (10–13). Similarly, ExPEC is a leading cause of urinary tract infection in dogs and cats (14–16) and causes a range of extraintestinal diseases in companion animals in general (17, 18). Many studies of ExPEC in companion animals have focused on isolates with resistance to clinically important antimicrobials. For example, a recent study in New Zealand showed that humans living in a household shared the same drug-resistant *E. coli* with their pet dogs (19). Although these studies are clearly important, narrowing the scope of research to only antimicrobial-resistant isolates could obscure a deeper understanding of the epidemiology of *E. coli* sequence types associated with clinical disease in companion animals.

Despite its vast commensal and pathogenic range, deciphering the zoonotic and zooanthroponotic potential of *E. coli* remains a challenge (20, 21). Addressing this issue requires a deep understanding of genomic epidemiology of the dominant ExPEC sequence types (STs) as well as emerging ExPEC STs in both humans and animals (7, 22–26). There are more than 11,000 *E. coli* sequence types, but the top 20 ExPEC sequence types are responsible for more than 85% of ExPEC infections in humans (13). The remaining 15% of infections are caused by strains that display remarkable diversity, a pool that presumably harbors both clones which might emerge as novel pandemic lineages and others that might be less successful. However, the ability to predict which clone is which, and where the true reservoirs lie, currently remains outside our understanding.

There is an already-noted overlap in the STs that cause ExPEC infections in humans and dogs, with both species sharing ST73, ST131, and ST12 among the dominant types (8, 13). Among ExPEC infections in dogs, phylogroup B2 *E. coli* is highly prevalent, with ST372, ST73, ST127, ST12, and ST131 (8), as well as drug-resistant *E. coli* in commensal phylogroup A, with ST410 and ST683 the most common (18).

The role of plasmids in the evolution of ExPEC STs, their diversification into sublineages, and dissemination in nonhuman niches is an emerging theme in the genomic epidemiology of ExPEC. Despite the variety of plasmids that are found in *E. coli*, F plasmids dominate among a proportion of major and emerging ExPEC STs. For example, carriage of F plasmids is very common in ST131, ST95, ST58, and ST127, with separate plasmid lineages found to be characteristic of ST sublineages (7, 22, 23, 25). F plasmids can be categorized genotypically by their accessory gene content, and plasmid lineages within those genotypes can be approximated by their F replicon sequence types (RSTs). In terms of the categorical distinction, two major F plasmid genotypes, namely, ColV plasmids and ColIa/pUTI89-like plasmids, are commonly found in ExPEC (27, 28). Both types carry genetically distinct, but functionally similar, accessory gene loci that are primarily involved in iron acquisition. Iron acquisition genes function as both intestinal fitness factors and extraintestinal virulence factors, underscoring their obvious utility to *E. coli* (29, 30). In contrast to pUTI89-like plasmids, which rarely carry antimicrobial resistance genes (ARGs), ColV plasmids often carry ARG loci in association with smaller mobile genetic elements (MGEs), such as insertion sequences and transposons (28). Currently, little is known of their distribution within ExPEC in dogs.

Here, we have undertaken the largest whole-genome sequencing analysis of *E. coli* from dogs. The collection comprises 377 isolates, collected over 11 years from a restricted geographical area. Most isolates were from cases of extraintestinal diseases, such as urinary tract, respiratory, and skin infections. A small number of isolates from gastrointestinal tract

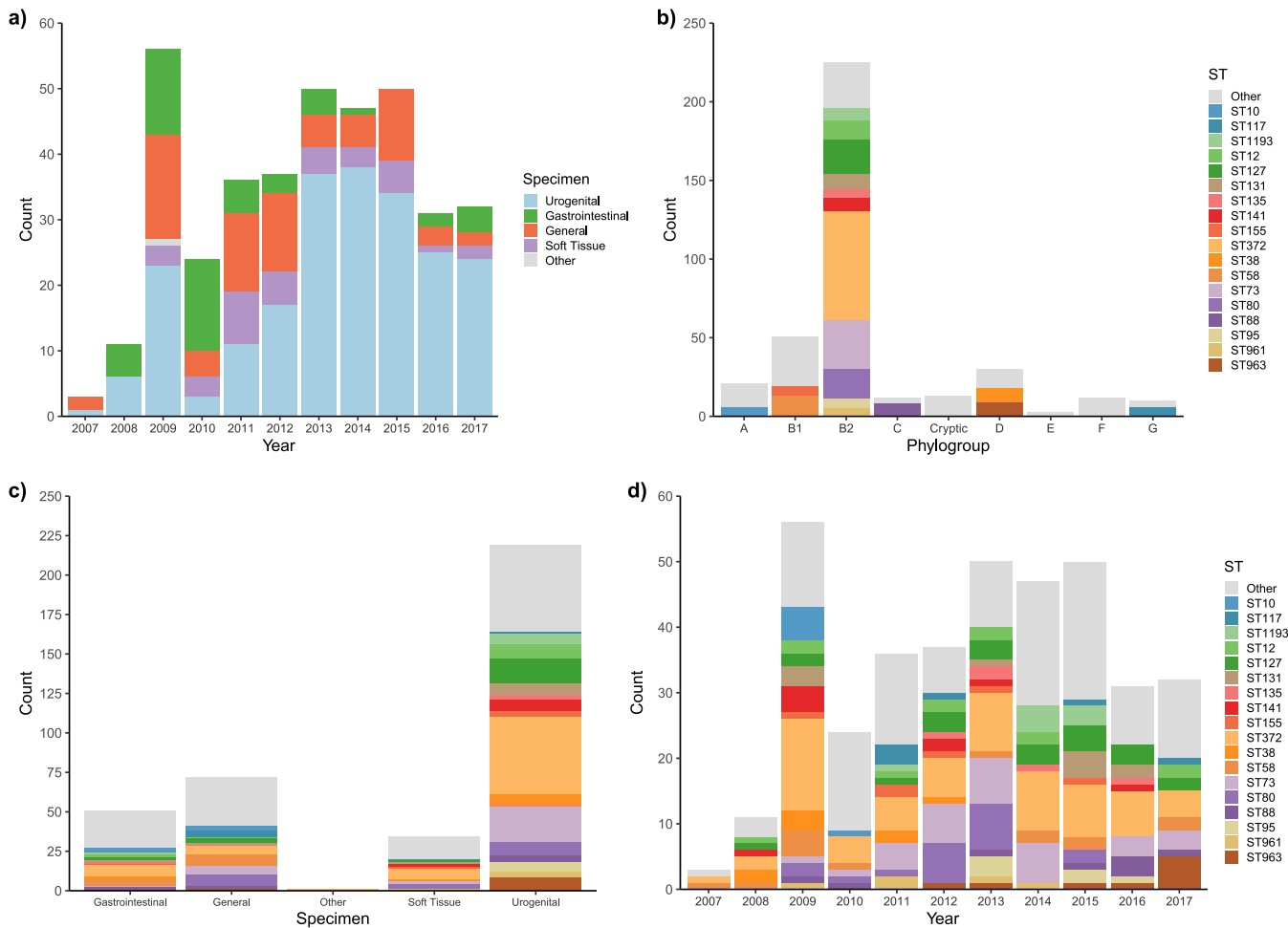

**FIG 1** Characteristics of the genome collection. (a) Counts of isolates by year of isolation, stratified by specimen type. (b) Counts of phylogroups, stratified by ST. (c) Counts of specimen type, stratified by ST. (d) Counts of STs identified for each year of isolation.

specimens are also represented in the collection. We have determined their phylogroups, multilocus sequence types, e-serotypes, and *fimH* types. Furthermore, we have defined their plasmid repertoire, noting carriage of important F plasmids, and we carried out a thorough analysis of their virulence-associated gene (VAG) and antimicrobial resistance gene carriage.

## RESULTS

**Study collection.** The study collection consisted of 377 *E. coli* genome sequences originating from dogs presenting at veterinary hospitals in Melbourne, Australia. *E. coli* was isolated over an 11-year period from 2007 to 2017 (Fig. 1a). Most isolates were of extraintestinal origin, and specimens were classified as urogenital tract (219, 58.1%), general (72, 19.1%), soft tissue (34, 9%), and other (one isolate had conflicting source information) (Fig. 1a). Fifty-one gastrointestinal tract isolates (13.5%), primarily from bile or feces, were also included.

B2 was the dominant phylogroup (225, 59.7%), followed by B1 (51, 13.5%), D (30, 8.0%), and A (21, 5.6%) (Fig. 1b). A total of 53 distinct STs were identified, and 18 of these had five or more representatives. STs with less than 5 representatives and isolates for which an ST could not be identified were grouped as "other." The five most prevalent STs were ST372 (69, 18.3%), ST73 (31, 8.2%), ST127 (22, 5.8%), ST80 (19, 5.0%), and ST58 (14, 3.7%). Apart from ST58 (B1), these STs all belonged to phylogroup B2. Other well-recognized ExPEC STs identified included ST12, ST131, ST141, ST963, ST1193, ST95, ST88, and ST38. The prominent commensal ST10 and the avian-associated ST117 were also identified.

Among the 18 major STs, 17 were present in urogenital tract specimens, while 12 were present in gastrointestinal tract specimens. While extraintestinal isolates were presumptively

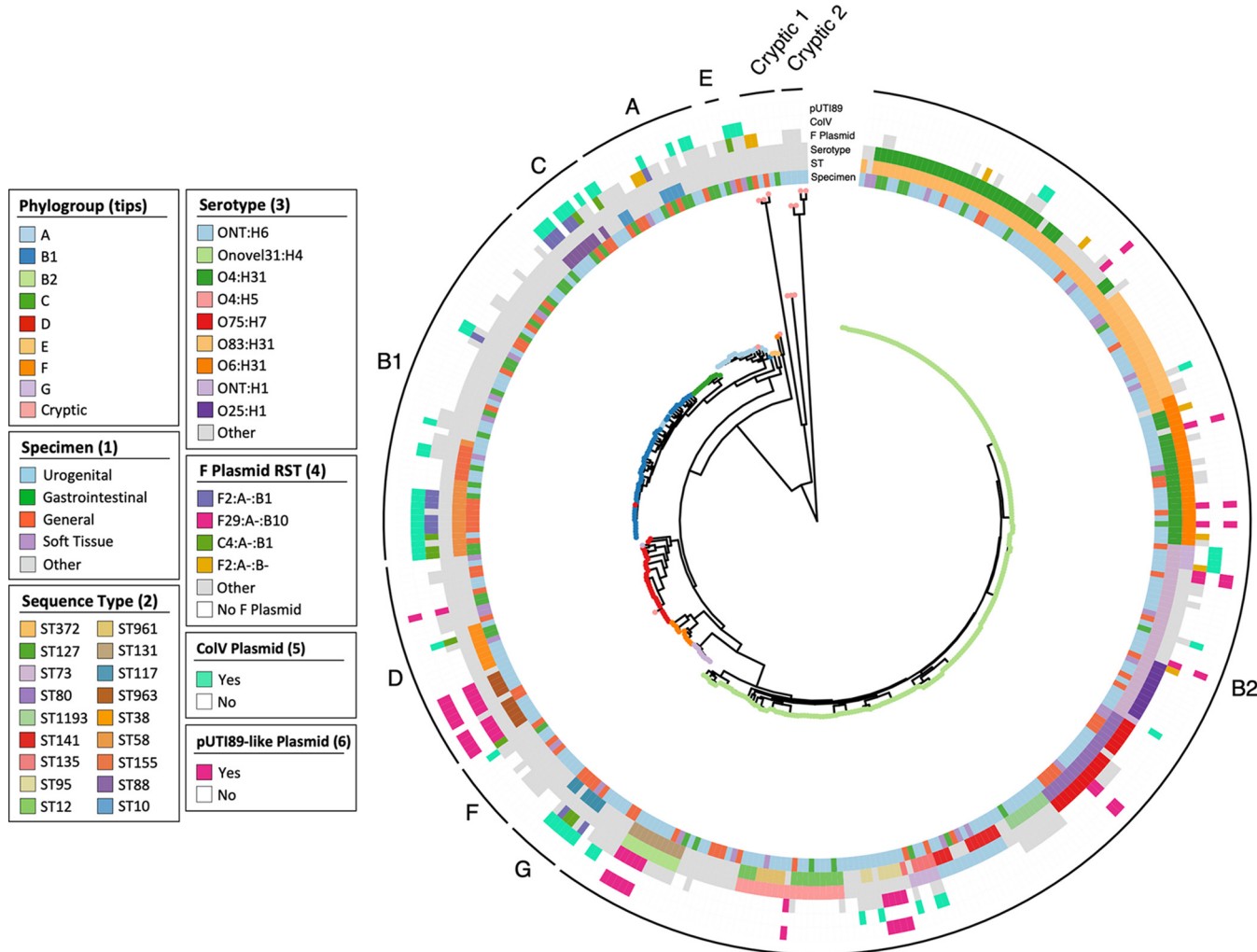

**FIG 2** Core gene phylogeny, as shown by a maximum-likelihood phylogeny inferred by IQTree on the core gene alignment produced by Roary. Clades are labeled on the outermost ring by consensus phylogroup (determined by EZCLermont), shown on tree tips. Metadata for specimen, ST, serotype, F plasmid RST, ColV plasmid and pUTI89 plasmid presence are represented in bands around the phylogeny.

the etiological agent responsible for presentation, it is possible that a proportion of gastrointestinal tract isolates were actually commensals, not intestinal pathogens. The facts that (i) ExPEC typically originates in the lower gastrointestinal tract, (ii) the ST distribution in gastrointestinal tract samples comprised most of the dominant urinary STs, and (iii) there was a lack of classical intestinal virulence factors among gastrointestinal tract isolates suggest that most gastrointestinal tract samples in the collection were intestinal commensals with the potential to cause extraintestinal disease.

Temporal persistence of several major STs, including ST372 (observed every year), ST127 (9 of 11 years), ST58 and ST73 (both 8 of 11 years), was detected (Fig. 1d). Only ST10 was seen in fewer than three sampling years. Despite the variable sample size each year, these data generally indicate that certain lineages of canine pathogenic *E. coli* are consistently present in the population over an extended period of time.

**Phylogeny.** A maximum-likelihood core gene phylogeny was clearly structured by phylogroup and ST, as expected (Fig. 2). It did, however, reveal that the *in silico* phylogroup, as determined by EZClermont, was occasionally inaccurate, as illustrated by the presence of several sequences designated phylogroup A in the B1 clade, as well as two miscalls in the D clade. Overlaid metadata did not reveal any clustering of specimen by phylogeny, reflecting the previously described presence of STs across specimen types. Specific serotypes generally corresponded to STs, but O4:H5 was found in both ST12 and ST961.

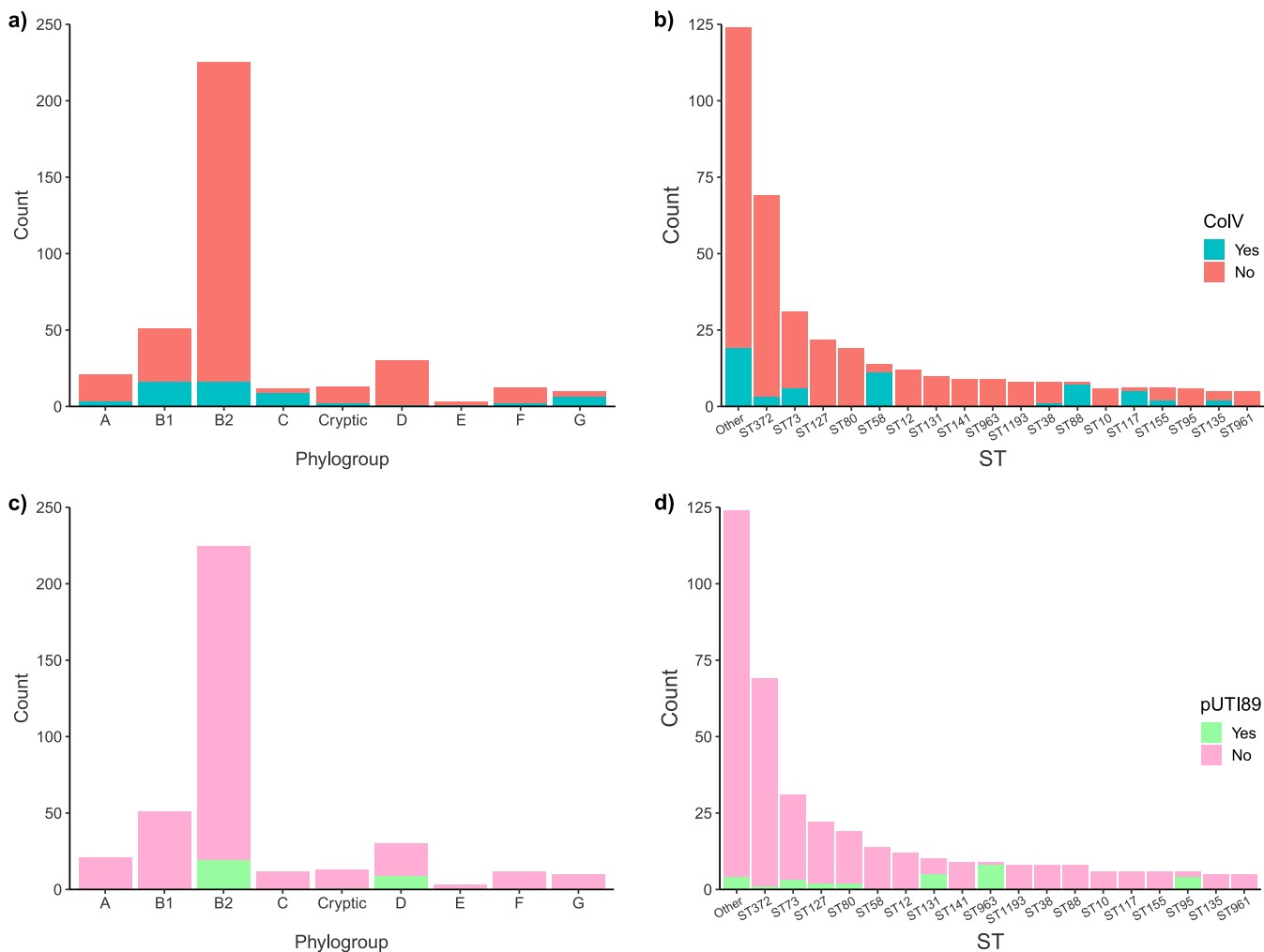

**FIG 3** F plasmid carriage. Inference of ColV plasmid carriage based on phylogroup (a) and ST (b) and of pUTI89-like plasmid carriage based on phylogroup (c) and ST (d).

**F plasmids.** Important F plasmid archetypes ColV (56/377, 14.9%) and pUTI89-like (29/377, 7.7%) were identified in a proportion of isolates. ColV$^+$ sequences carried a variety of F RSTs including A-:B1:C4, F2:A-:B1, and F2:A-:B-, while all pUTI89$^+$ sequences carried the F29:A-:B10 replicon (Fig. 2). These two plasmid types had different phylogenetic distributions. ColV plasmids were found in eight major STs belonging to all phylogroups (Fig. 3a and b), whereas pUTI89-like plasmids were present in seven STs restricted to phylogroups B2 and D (one pUTI89$^+$ sequence was typed as cryptic but belonged to phylogroup D according to the phylogenetic tree, indicating that it was not a true member of the cryptic clades). ST372 and ST73 were the only STs that contained both ColV$^+$ and pUTI89$^+$ sequences, but most members of these STs carried neither plasmid. Consistent with principles of plasmid exclusion, no sequence was found to contain both plasmid types.

**Antimicrobial resistance.** Antimicrobial resistance (AMR) to seven antimicrobial compounds comprising six drug classes commonly used in veterinary medicine was tested for by disk diffusion for 365/377 isolates (see Fig. S1 in the supplemental material). Antimicrobials tested included ampicillin, amoxicillin plus clavulanate, cephalexin, enrofloxacin, tetracycline, sulfamethoxazole, and trimethoprim. On average, isolates were only resistant to 1.26 compounds, with a median of zero. Seventy-seven isolates (20.4%) were resistant to three or more antimicrobial classes and were therefore considered multidrug resistant.

The class 1 integrase gene *intI1*, a common genetic proxy for multidrug resistance, was found in 51 sequences (13.5% at 90% sequence identity over 95% of the length of *intI1*), 40 of which were multidrug resistant (MDR). Truncated copies of *intI1* were also detected, the most common of which was a 746-bp fragment in ST1193 (7/8, 87.5%) and ST131 (3/10, 30%). A

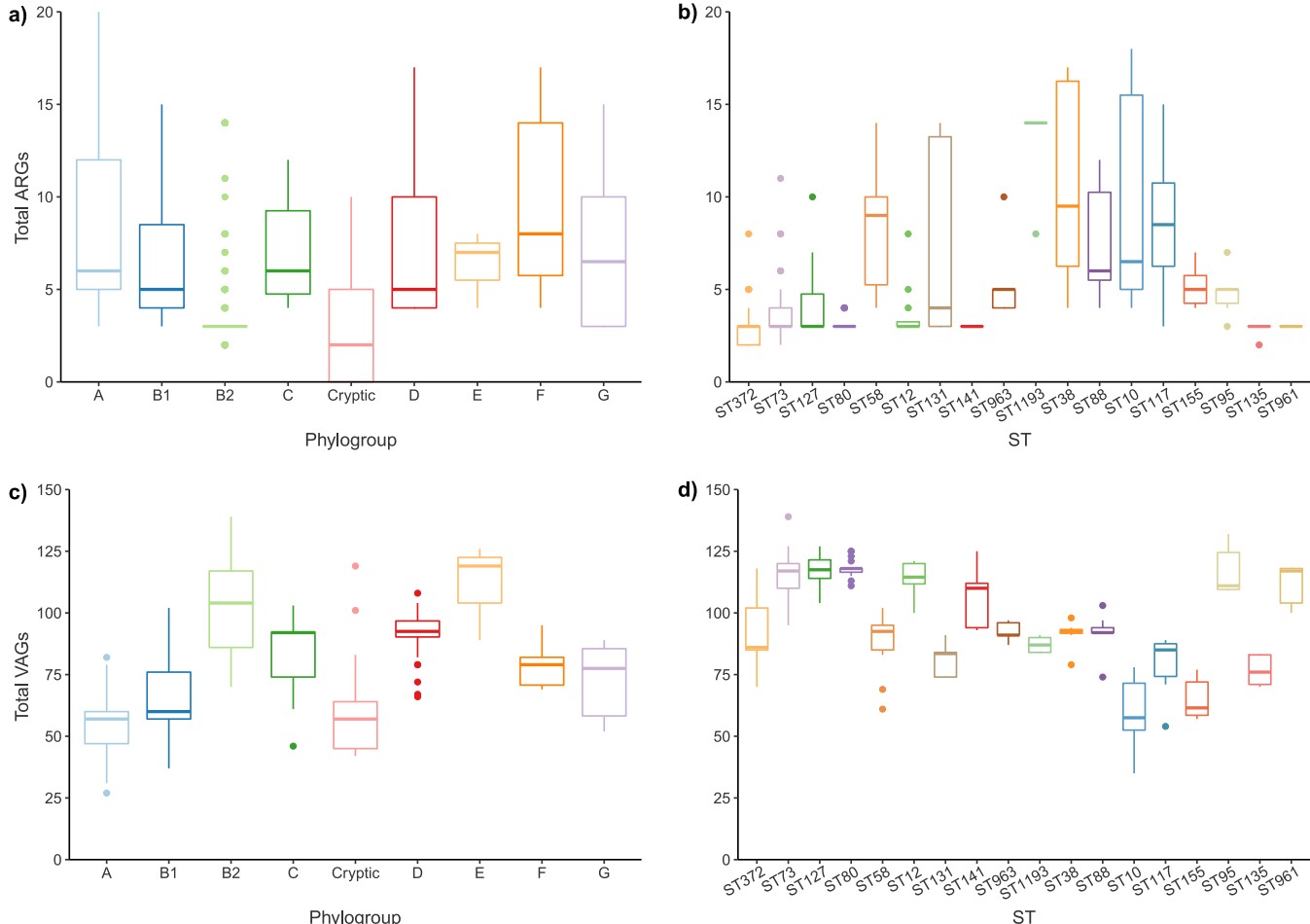

**FIG 4** Distribution of antimicrobial resistance genes (ARGs) and virulence-associated genes (VAGs). Boxplots show average total ARGs by phylogroup (a), average total ARGs by ST (b), average total VAGs by phylogroup (c), and average total VAGs by ST (d).

further 10 isolates carried truncated copies of *intI1* ranging in size from 114 to 746 bp, underscoring the ongoing evolution of the class 1 integron structure. All 20 sequences carrying truncated copies of *intI1* were classified as MDR, and 2 of them (MVC207 and MVC227) also carried full copies of *intI1* on other contigs. The total inferred carriage of class 1 integrons was therefore 70/377 (18.6%).

Genotypic AMR in the collection was generally moderate and reflective of the phenotypic data, with a mean 5.12 ARGs per sequence and a median of 4. Phylogroup B2 sequences displayed a low prevalence of ARGs (mean, 3.89), second only to cryptic phylogroup (mean, 3.15) (Fig. 4a; see also Fig. S1 in the supplemental material). Predominant phylogroups B1 (mean, 6.35) and D (mean, 7.8) showed moderate carriage, while phylogroup F, to which only 12 sequences belonged, showed the highest prevalence (mean, 9). Common ARGs included $bla_{\text{TEM-181}}$ (98, 26%), *sul2* (59, 15.6%), *strA* (*aph-3-lb*; 54, 14.3%), *strB* (*aph-6-ld*; 51, 13.5%), *tetA* (47, 12.5%), and *sul1* (44, 11.7%).

The five most resistant STs in terms of the number of ARGs they contained were B2-ST1193 (mean, 13.2), D-ST38 (mean, 10.5), A-ST10 (mean, 9.67), G-ST117 (mean, 8.67), and B1-ST58 (mean, 8.29) (Fig. 4b). ST1193 was conspicuous as the only ST within phylogroup B2 to have consistently high rates of ARG carriage. All five STs with high ARG carriage showed evidence of integron carriage, including presence of characteristic components *intI1*, ARG cassettes (*dfrA* and *aadA* variants), and *qacEΔ1*. Within ST58 and ST117, *intI1* carriage often co-occurred with ColV plasmid carriage.

Fluoroquinolone resistance mediated by point mutations was predicted for 66 sequences (17.5%), only 33 of which were phenotypically resistant to enrofloxacin. All ST131 (10) and

ST1193 (8) sequences were predicted to be fluoroquinolone resistant. Other notable STs predicted to be resistant included ST38 (5/8), ST58 (2/14), ST73 (2/31), ST127 (2/22), and ST10 (2/6).

**Virulence.** In contrast to AMR genotypes, VAG profiles were extensive—consistent with the pathogenic status of the isolates under investigation. The dominant phylogroup B2 had the second highest average VAG count, with 102 VAGs per strain (Fig. 4c). Phylogroup E had the highest average VAGs (mean, 111 per strain), but only contained three sequences (MVC307, MVC681, and MVC73). Phylogroups D and C also had extensive VAG arrays, with averages of 90.9 and 83.2 per strain, respectively. The sequence types with the most virulence genes (mean VAGs are in parentheses) within phylogroup B2 were ST80 (118), ST127 (118), ST95 (117), ST73 (116), ST12 (114), ST961 (111), and ST141 (107) (Fig. 4d). Sequences belonging to ST1193 (87.1) and ST372 (89.1) carried fewer VAGs than their B2 counterparts, a notable observation given the extensive ARG carriage of ST1193 and the dominance of ST372 in the collection. STs with high VAG carriage in other phylogroups included ST963 and ST38 (phylogroup D; means of 92.7 and 91.5, respectively) and ST88 (phylogroup C; 91.9). Generally, STs with higher ARG carriage displayed more moderate VAG carriage.

VAGs in the collection overwhelmingly encoded functions associated with ExPEC. Genes associated with intestinal pathotypes were very rare, with only two sequences containing a toxin gene of any kind (MVC147-ST10 for *estIa* and MVC785-ST2700 for *toxB*), and Shiga toxin genes were not detected in any sequence (see Table S1 in the supplemental material).

As B2 sequences carried high average VAGs, numerous specific genes were predictably clustered within this phylogroup (see Fig. S2 in the supplemental material). The hemolysin operon *hlyABCD* was mostly found in B2 sequences (total *hylA* of 144, 38.2%). The heme uptake operon *chuASTUVWXY* was present in all B2 sequences except for a single sequence missing *chuA* (total of 227, 60.2%). The K1 capsule genes (kpsCDEFMSU) were typically found as an operon in B2, G, F, D, and cryptic clades, whereas phylogroups B1, A, and C lacked these genes (total *kpsM* of 185, 49%). Most ST372 sequences lacked *kps* genes, a rarity among B2 sequences. P fimbriae encoded by genes of the *pap* operon were primarily found in B2 sequences with variability in the presence/absence patterns of individual gene carriage (total *papC* of 150, 39.8%). F1C fimbriae were only found in B2 sequences (ST372 and ST73), except for a single B1-ST155 sequence (total *focA* of 99, 26.3%). Genes of the *sfa* operon encoding S fimbriae were variably present, mostly within B2 sequences, though major subunit *sfaA* was mostly absent (13, 3.5%; total *sfaB* of 179, 47.5%).

Genes involved in iron acquisition that are typically found on ColV plasmids, such as *iroN* (220, 58.4%), *iucD* (78, 20.7%), and *iutA* (77, 20.4%), were also identified in sequences that did not carry a ColV plasmid, indicating chromosomal locations or carriage on other episomal elements. In B2, *iroN* carriage did not usually correspond to ColV plasmid carriage (B2: 12 ColV$^+$ versus 167 ColV$^-$), whereas in non-B2 sequences, *iroN* was almost always observed in conjunction with ColV (non-B2: 39 ColV$^+$ versus 2 ColV$^-$). Similarly, *cjrABC-senB*, a component of pUTI89-like plasmids and putative iron uptake system, was identified in sequences with and without pUTI89-like plasmids (29 pUTI89$^+$, 25 pUTI89$^-$; total of 54 *cjrABC-senB*, 14.3%), although these genes were restricted to phylogroups B2 and D regardless of plasmid carriage. Overall, these results point to an apparent interplay between specific iron acquisition systems, mobile genetic elements, and phylogenetic background within pathogenic *E. coli*. Further underscoring the importance of iron acquisition in pathogenic *E. coli* was the extensive carriage of the yersiniabactin high-pathogenicity island (HPI), indicated by marker genes *fyuA* and *irp2* (both 287, 76.1%). HPI was present in almost all B2 sequences (222/225, 98.7%) and identified in every phylogroup except for E.

## DISCUSSION

**Canine and human ExPEC share common lineages.** Here, we generated the largest collection of canine-origin *E. coli* whole-genome sequences assembled to date. Phylogroup B2 dominated the collection. Phylogroup B2 was similarly overrepresented (79.6%) in a study of 618 *E. coli* isolates from dogs attending four veterinary clinics in France (18) and is by far the major phylogroup among human ExPEC (13, 31). A diversity of STs were identified across the specimen types, indicating that STs are not syndrome specific. The most common STs almost

exclusively (barring the enigmatic B1-ST58) belonged to phylogroup B2, with ST372, ST73, and ST127 being the most prevalent types.

The most common type, ST372, was similarly dominant among *E. coli* from canine infections in Australia and fecal commensals from healthy dogs in Spain (8, 18, 32, 33). ST372 is also identified among human ExPEC isolates, although, unlike ST73 and ST127, it is not ranked in the top 20 most prevalent human ExPEC STs (34, 35). It was interesting that the virulence gene carriage of ST372 was lower than that of many of the other STs in the collection. This might be explained by database bias toward genes associated with virulence for humans and a lack of knowledge of dog-specific virulence genes. Alternatively, the dominance of the ST in the absence of an extensive array of virulence genes may be due to its possession of metabolic capacities facilitating success in the canine gut, with its infectivity being predominantly host-mediated. The latter possibility, in conjunction with its apparent dominance in both healthy and diseased dogs and lower prevalence in humans, supports the contention that ST372 is a dog-adapted lineage of *E. coli* (18). Future work involving in-depth genomic comparisons between dog and human ST372 isolates will provide further information in this regard.

ST73 is often overshadowed by ST131, despite consistent reports of its dominance as a human ExPEC sequence type causing UTI and bacteremia in Australia (36, 37), France (38), the United Kingdom (39, 40), and elsewhere (13). The dominance of ST73 in our study of *E. coli* from dogs, as well as in the large study of isolates from dogs in France, indicates that dogs may be a major reservoir of ST73 (18). A similar observation prevails in *E. coli* isolated from cats (41). It is as yet unknown whether the ST73 in dogs represents distinct dog-adapted sublineages or acquisition of human lineages, as has been postulated for ST131 and ST1193 (42).

ST127 is another common human ExPEC lineage, often associated with sepsis (43). We recently demonstrated that there is global-scale genomic linkage between ST127 from companion animals (including from this collection) and human origin ST127, reflecting the aforementioned scenario with ST131 and ST1193 (7, 42).

Apart from ST372, most of the common STs in this collection are well-described human ExPEC, and their presence in community-onset canine infections over multiple years suggests that dogs and humans are both colonized by a broad spectrum of human ExPEC lineages. This is easily explained by their close association with humans as companion animals, with cocarriage being driven by shared living spaces, physical proximity, and consumption of overlapping diets, which can include raw retail meats and human food scraps. Consistent with this view, these *E. coli* STs predominate in the feces of healthy humans (44). Further supporting the sharing of ExPEC with humans is the virulence content of major STs in the collection reflected that of their human counterparts, with a high prevalence of Yersinia HPI carriage and an abundance of other iron acquisition genes that function in both gut colonization and pathogenicity (29).

Overall, our results indicate that canine ExPEC mostly comprise commensals and pathogens that are commonly associated with humans. ST372 represents an exception in this regard; it is a lineage that may be mostly adapted to canines, with a lower prevalence in humans. What these results imply about relative rates and directions of transfer of ExPEC between humans and dogs is still uncertain and requires further investigation.

**Major F plasmid types circulate in canine ExPEC.** The major human ExPEC STs, particularly ST95 (22), ST131 (23, 26), and ST73 (45), comprise sublineages that are often discernible by serotype and *fimH* allele variation. Lineage subdivision is also often accompanied by carriage of different F plasmid replicon sequence types belonging to ColV (various replicon types) and pUTI89-like (ColIa$^+$/F29:A-:B10) genotypes (22, 25, 26). In the canine collection, 56 isolates (14.9%) carried a ColV plasmid and these sequences were identified across all phylogroups. ExPEC ColV carriage in our canine collection was similar to the estimated human ExPEC carriage rate of 16% (25). ColV plasmids were particularly dominant in phylogroups C (ST88) and G (ST117), but a significant proportion of phylogroup B1 (ST58) and a small number of B2 (numerous STs) isolates also carried a ColV plasmid. ST88 is a major ColV plasmid-carrying ST causing extraintestinal disease and has also been identified in store-bought produce (36, 46–48). ST58 is a multihost pathogen with a major sublineage rich in ColV plasmids and

is frequently identified in poultry and pigs (25). ST117 is a noted avian pathogenic *E. coli* (APEC) lineage rich in ColV plasmids and is dominant in both commensal and pathogenic *E. coli* populations from poultry in Australia and abroad (49, 50). As has been demonstrated for *Campylobacter* species, the frequent consumption of raw chicken by dogs may present a risk for acquisition of poultry-associated ExPEC, such as ST117 (51). The association of ColV-carrying ST58, ST88, ST131, and ST95 isolates with bacteremia or sepsis in Australia is also notable (36, 52). Our results are therefore reflective of previous reports of ColV carriage within prominent and emerging STs, demonstrate their pathogenicity in dogs, and support interspecies transfer between numerous hosts. It will be important to monitor the frequency of isolation of STs that can carry ColV plasmids, given the important role they play in *E. coli* that cause extraintestinal disease in humans and domestic animals (53).

In contrast to ColV plasmid carriage, only about 7.7% (29/377) of the collection carried a pUTI89-like plasmid. These plasmids are common in human ExPEC infections and typically carry *cjrABC-senB* virulence genes, which are purported to contribute to iron acquisition and ExPEC virulence in murine models of UTI (27, 54). pUTI89-like plasmids were confined to phylogroups B2 and D and were most common in ST963 (8/9, 89%), ST95 (3/6, 66.7%), and ST131 (5/8, 62.5%) sequences. Their presence in these STs is indicative of sublineage partitioning, as has been recently described for ST131 and ST95 (22, 23).

Despite the relevance of ColV and pUTI89-like plasmids to several important ExPEC STs, the two dominant STs in the collection, ST372 and ST73, mostly lacked F plasmids of these types, reiterating the unavoidable importance of the core genomic background in ExPEC evolution. Interestingly, ST73 was one of the most VAG-rich STs, with an average 116 VAGs, whereas ST372 carried substantially fewer, at an average of 89.1. ST73 isolates in this collection and others were shown to carry an abundance of adhesins and genes involved in iron acquisition, suggesting a redundancy for genes of ColV or pUTI89-like plasmids (45). Despite the lower total number of VAGs, ST372 in this collection also carried adhesins and iron acquisition genes. Overall, this indicates that our understanding of the relative contribution of plasmids and chromosomally encoded gene functions to ExPEC intestinal fitness and extraintestinal pathogenicity is still in its infancy.

**Antimicrobial resistance carriage is low but not trivial.** Notably, carriage of clinically important antibiotic resistance genes was not a common occurrence in the collection, but persistent drug resistant lineages were seen and were associated with class 1 integron carriage, often in the presence of ColV plasmids. Class 1 integrons with complete or truncated copies of *intI1* were identified in 70/377 (18.6%) *E. coli* isolates, 27 (38.6%) of which carried a ColV plasmid. Carriage of *intI1* with and without ColV was notable in several STs, including ST58 (ColV$^+$), ST117 (ColV$^+$), ST10 (ColV$^{+/-}$), and ST38 (ColV$^-$). The presence of *intI1*$^+$ ARG loci on ColV plasmids has been well described, and our results indicate that this trend extends to canine ExPEC (28, 52, 55, 56). Extended-spectrum beta-lactamase (ESBL)-producing *E. coli* was infrequently detected; however, ESBL-producing *E. coli* in ST131, ST1193, and AmpC $\beta$-lactamase-producing *E. coli* in ST155, ST315, ST617, ST457, ST767, and ST372 have been detected in companion animals around the world (32, 57, 58). Fluoroquinolone resistance mutations were detected in 17.5% of sequences, also indicating exposure to human sources of *E. coli*, with successful human ExPEC lineages ST131 and ST1193 presenting as the predominant carriers of these mutations (57, 59).

**Conclusion.** In summary, dogs are primarily infected with ExPEC characteristic of human infections, as characterized by the distribution of STs, phylogroups, plasmids, AMR, and virulence genes. One notable exception is ST372, which may be a "dog-adapted" ST that has spilled back into humans. The overall explanation for the shared genotypes is shared gut carriage of these *E. coli* between humans and their pets due to close proximity and shared diets.

## MATERIALS AND METHODS

**Isolates used in this study.** The 377 *E. coli* isolates analyzed were part of a collection isolated from dogs presenting with various extraintestinal diseases by the Clinical Microbiology Laboratory of the Melbourne Veterinary School, University of Melbourne, Australia, between 2007 and 2017. The isolates were transported as slope cultures on LB agar. Isolate names carry the prefix MVC (for Melbourne Veterinary

Collection) followed by a 1- to 3-digit number specifying individual isolates from the collection. Full isolate metadata and public accession numbers are available in Table S1 in the supplemental material.

**Genomic DNA isolation, whole-genome sequencing, and assembly.** *E. coli* isolates from the Melbourne Veterinary Collection were freshly cultured onto LB agar plates, and a single colony was used to inoculate 5 mL of sterile LB medium. Following overnight culture, total cellular DNA was extracted using the ISOLATE II Genomic DNA (Bioline) kit following the manufacturer's standard protocol for bacterial cells and was stored at 4°C. Library preparation was done by the AIMI Core Sequencing Facility, University of Technology Sydney, following the adapted Nextera Flex library preparation kit process Hackflex (60). Briefly, genomic DNA was quantitatively assessed using the Quant-iT PicoGreen dsDNA assay kit (Invitrogen, USA). Each sample was normalized to a concentration of 1 ng/$\mu$L. A 10-ng sample of DNA was used for library preparation. After tagmentation, DNA was amplified using the facility's custom-designed i7 and i5 barcodes, with 12 cycles of PCR. Due to the number of samples, the quality control for the samples was done by sequencing a pool of samples using the MiSeq V2 Nano kit for 300 cycles. Briefly, after library amplification, a 3-$\mu$L sample of each library was added to a library pool. The pool was then cleaned up using SPRIselect beads (Beckman Coulter, USA) following the Hackflex protocol. The pool was sequenced using the MiSeq V2 nano kit (Illumina, USA). Based on the sequencing data generated, the read count for each sample was used to identify the failed libraries (i.e., libraries with less than 100 reads) and normalized to ensure equal representation in the final pool. The final pool was sequenced on one lane of an Illumina Novaseq S4 flow cell, 2 × 150 bp, at Novogene (Singapore). The quality of reads generated was confirmed with fastp (0.20.1).

**Genome assembly and gene screening.** Clermont phylogrouping was performed with EZCLermont (https://github.com/nickp60/EzClermont). A modular analysis pipeline known as pipelord2, implemented with the Snakemake workflow management system, was used to perform primary bioinformatic analysis (61). This pipeline is freely available to download from https://github.com/maxlcummins/pipelord2_0. Default settings were used unless otherwise stated. First, Kraken2 was applied to the sequence reads to confirm all genomes were *E. coli*. Draft genomes were then assembled with Shovill 1.0.4 (https://github.com/tseemann/shovill), with default settings and assembly stats run to confirm the quality of the assemblies (https://github.com/sanger-pathogens/assembly-stats). Assemblies with >800 contigs or total lengths of <4.5Mbp or >6.5Mbp were excluded. MLST 2.19.0 (https://github.com/tseemann/mlst) was used to determine *E. coli* sequence types. ABRicate 1.0.1 (https://github.com/tseemann/abricate) was used to screen draft genomes for genes from several publicly available and custom in-house databases. Public databases used were CARD, VFDB, PlasmidFinder, SerotypeFinder, and ISFinder (62–66). The custom database included the set of genes used to infer ColV plasmid carriage (see below) and additional virulence genes. This is available at https://github.com/maxlcummins/custom_DBs. ABRicate was also used to align assemblies to a variety of reference plasmids, including pUTI89 from the *E. coli* strain UTI89, sourced from GenBank (gb | NC_007941). The pMLST tool available at https://bitbucket.org/genomicepidemiology/cge-tools-docker/src/master/ was used to perform pMLST typing (67). AMR-associated single-nucleotide polymorphisms were identified with PointFinder (68). Finally, gene screening results were summarized using abricateR (https://github.com/maxlcummins/abricateR), with a gene being considered present at 95% length and 90% nucleotide identity.

**Criteria for inference of plasmid presence.** The presence of a ColV-type plasmid was inferred using criteria previously described by Liu et al. (69). The presence of a pUTI89-like plasmid was inferred if a given assembly mapped to ≥90% of the pUTI89 sequence at ≥90% identity or if the isolate was determined by pMLST to carry the F29:A-:B10 RST combination, which is characteristic of pUTI89-like plasmids.

**Pan-genome and phylogenetic analysis.** The assembled genomes were annotated using prokka 1.14.6 (70). The core and pan-genome were then determined with Roary 3.13.0, with default settings and paralog splitting on (71). The resulting core gene alignment of 1,770,948 bp was then used as the basis for subsequent analyses. IQTree 2.0.3 was used to infer a maximum-likelihood phylogenetic tree using the GTR+F+R substitution model and 1,000 bootstrap replicates (72). The tree was midpoint rooted for visualization.

**Data analysis and visualization.** A custom R script was written in RStudio 1.4.1106 with R 4.0.5 to analyze and visualize the data generated by pipelord, including MLSTs, ARGs, VAGs, and MGEs. This script was also used to infer the presence of plasmids, based on BLAST data generated by pipelord with plasmidmapR (https://github.com/maxlcummins/plasmidmapR), and to visualize the phylogenetic tree in conjunction with metadata and gene data. The sequences of plasmids pCERC4 and pUTI89 were visualized with SnapGene Viewer (version 5.0.7; GSL Biotech LLC). Microsoft PowerPoint was used to compile elements of Fig. 2 and Fig. S1 to S5 in the supplemental material.

**Data availability.** All genomes used in this study were deposited in GenBank and the Sequence Read Archive under the BioProject PRJNA678027. Individual accession numbers can be found with comprehensive metadata and genomic data in Table S1. The data analysis and visualization script are freely available at https://github.com/CJREID/MVC and can be used to reproduce all secondary analyses. R package versions used therein are available within the README.md document in the code repository.

## SUPPLEMENTAL MATERIAL

Supplemental material is available online only.
**SUPPLEMENTAL FILE 1**, PDF file, 5.6 MB.
**SUPPLEMENTAL FILE 2**, CSV file, 0.6 MB.

## ACKNOWLEDGMENTS

We thank Kay Anantanawat for assistance with whole-genome sequencing. This research was supported by the Australian Government Research Training Program and via funding from The Australian Centre for Genomic Epidemiological Microbiology (AusGEM), a collaborative

partnership between NSW DPI and The University of Technology Sydney. Computing infrastructure was provided by the UTS Interactive High Performance Computing facility.

P.E. contributed formal analysis, investigation, data curation, and writing of the original draft; M.L.C. contributed investigation, data curation, project administration, and methodology; G.F.B. contributed investigation, data curation, and project administration; M.S.M. contributed investigation, data curation, and project administration; C.J.R. contributed conceptualization, methodology, software, validation, formal analysis, writing review and editing, visualization, and supervision; S.P.D. contributed conceptualization, resources, writing review and editing, supervision, project administration, and funding acquisition.

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
