## [Reviewer comments · Microbiology Spectrum]

Microbiology Spectrum

Genomic and temporal trends in canine ExPEC reflect those of human ExPEC

Paarthiphan Elankumaran, Max Cummins, Glenn Browning, Marc Marends, Cameron Reid, and Steven Djordjevic

Corresponding Author(s): Steven Djordjevic, Australian Institute for Microbiology and Infection

Review Timeline:

Submission Date:	April 7, 2022
Editorial Decision:	May 16, 2022
Revision Received:	May 22, 2022
Accepted:	May 24, 2022

Editor: Cheryl Andam

Reviewer(s): Disclosure of reviewer identity is with reference to reviewer comments included in decision letter(s). The following individuals involved in review of your submission have agreed to reveal their identity: Jorge Blanco (Reviewer #1); Tim Downing (Reviewer #2)

Transaction Report:

DOI: <https://doi.org/10.1128/spectrum.01291-22>

May 16, 2022

Prof. Steven Philip Djordjevic
Australian Institute for Microbiology and Infection
15 Broadway, University of Technology Sydney
Ultimo, NSW 2007
Australia

Re: Spectrum01291-22 (Genomic and temporal trends in canine ExPEC reflect those of human ExPEC)

Dear Prof. Steven Philip Djordjevic:

Link Not Available

Sincerely,

Cheryl Andam

Journals Department
Reviewer comments:

Reviewer #2 (Comments for the Author):

This paper is very good. It examines ExPEC in companion animal samples over time (2007-17). Although it is from a precise region, limited work in this area has been completed on human-animal transmission patterns in this context. It provides a comprehensive overview of the STs, VF genes, AMR genes and plasmid profiles in this unique collection. It identifies human-pet ExPEC transmission which is important for the community and our understanding of One Health. The data is understandably heterogeneous, so this study provides a foundation for future work. The sequence data is publicly available, as is the code (as far as I can see, >1200 lines of code so I didn't run it all) - it's good to see Open Science principles being applied here.

Major comment:

Discussion "which we argue tentatively supports a paradigm of primarily human to dog transfer" - I don't see how you can justify this from limited observational data alone. I think this clause is inaccurate, and overstates the strength of evidence available. One would need to model relative rates of dog-dog, human-human, dog-human, and human-dog transfer to quantify this to infer real relative transfer rates.

Minor comments:

Figure 1d - the y-axis should be the count (like in b) not the proportion IMHO, because the varying sample sizes per annum distort the perceived rates for the reader.

Methods: "A custom R script was written in RStudio 1.4.1106 with R 4.0.5 to perform secondary analyses on the primary data generated by the aforementioned methods, and to generate publication figures" - can you clarify which 2y analyses and which methods, it is not 100% clear for the reader.

Other comments:

Roary was used, which is fine for this study, but the authors should note for the future that there are now superior methods (at least 4).

Staff Comments:

Preparing Revision Guidelines

Please return the manuscript within 60 days; if you cannot complete the modification within this time period, please contact me. If you do not wish to modify the manuscript and prefer to submit it to another journal, please notify me of your decision immediately so that the manuscript may be formally withdrawn from consideration by Microbiology Spectrum.

Response to Reviewer

We would like to kindly thank the reviewer for taking the time to review our manuscript. We particularly appreciate the acknowledgement of our efforts to make data and analysis scripts available as we believe this is critical to good scientific practice in computational research. We have responded to all comments below and incorporated changes as suggested.

Kind regards,

Steven Djordjevic

Reviewer comments

This paper is very good. It examines ExPEC in companion animal samples over time (2007-17). Although it is from a precise region, limited work in this area has been completed on human-animal transmission patterns in this context. It provides a comprehensive overview of the STs, VF genes, AMR genes and plasmid profiles in this unique collection. It identifies human-pet ExPEC transmission which is important for the community and our understanding of One Health. The data is understandably heterogeneous, so this study provides a foundation for future work. The sequence data is publicly available, as is the code (as far as I can see, >1200 lines of code so I didn't run it all) - it's good to see Open Science principles being applied here.

Major comment:

Discussion "which we argue tentatively supports a paradigm of primarily human to dog transfer" - I don't see how you can justify this from limited observational data alone. I think this clause is inaccurate, and overstates the strength of evidence available. One would need to model relative rates of dog-dog, human-human, dog-human, and human-dog transfer to quantify this to infer real relative transfer rates.

Response: We concede that this tentative argument is not conclusively supported by the data analysis in this paper. Rather, it reflects a combination of observations in our genomic epidemiological data from this paper and others, as well as the intuition that most dogs would have a far smaller potential network for microbial transfer than their owners, favouring a scenario where they are primarily exposed to and colonised by *E. coli* circulating in human networks. That said, we have rephrased this paragraph to reflect the uncertainty that the reviewer rightly highlights and acknowledged the need for further investigation.

Minor comments:

Figure 1d - the y-axis should be the count (like in b) not the proportion IMHO, because the varying sample sizes per annum distort the perceived rates for the reader.

Response: Changed figure to count and updated legend in manuscript file.

Methods: "A custom R script was written in RStudio 1.4.1106 with R 4.0.5 to perform secondary analyses on the primary data generated by the aforementioned methods, and to

generate publication figures" - can you clarify which 2y analyses and which methods, it is not 100% clear for the reader.

Response: Updated as requested.

Other comments:

Roary was used, which is fine for this study, but the authors should note for the future that there are now superior methods (at least 4).

Response: Fair point, we will be updating to Panaroo for future work!

May 24, 2022

Prof. Steven Philip Djordjevic
Australian Institute for Microbiology and Infection
15 Broadway, University of Technology Sydney
Ultimo, NSW 2007
Australia

Re: Spectrum01291-22R1 (Genomic and temporal trends in canine ExPEC reflect those of human ExPEC)

Dear Prof. Steven Philip Djordjevic:

Your manuscript has been accepted, and I am forwarding it to the ASM Journals Department for publication. You will be notified when your proofs are ready to be viewed.

Sincerely,

Cheryl Andam
Editor, Microbiology Spectrum

Journals Department
Supplemental Dataset: Accept
Supplemental Material 1: Accept